# Implementation of Whole Genome Sequencing of Tuberculosis Isolates in a Referral Center in Rome: Six Years’ Experience in Characterizing Drug-Resistant TB and Disease Transmission

**DOI:** 10.3390/antibiotics13020134

**Published:** 2024-01-30

**Authors:** Angela Cannas, Ornella Butera, Antonio Mazzarelli, Francesco Messina, Antonella Vulcano, Mario Pasquale Parracino, Gina Gualano, Fabrizio Palmieri, Antonino Di Caro, Carla Nisii, Carla Fontana, Enrico Girardi

**Affiliations:** 1National Institute for Infectious Diseases “Lazzaro Spallanzani”—IRCCS, 00149 Rome, Italy; angela.cannas@inmi.it (A.C.); ornella.butera@inmi.it (O.B.); antonio.mazzarelli@inmi.it (A.M.); francesco.messina@inmi.it (F.M.); antonella.vulcano@inmi.it (A.V.); mario.parracino@inmi.it (M.P.P.); gina.gualano@inmi.it (G.G.); fabrizio.palmieri@inmi.it (F.P.); carla.fontana@inmi.it (C.F.); enrico.girardi@inmi.it (E.G.); 2Department of Medicine, UniCamillus International University, 00131 Rome, Italy; antonino.dicaro@unicamillus.org

**Keywords:** whole genome sequencing, tuberculosis, drug resistance, infection control

## Abstract

Over the past years, Tuberculosis (TB) control strategies have been effective in reducing drug-resistant (DR) TB globally; however, a wider implementation of new diagnostic strategies, such as Whole genome sequencing (WGS), would be critical for further improvement. The aim of this study, based on WGS of *Mycobacterium tuberculosis* (MTB) strains isolated in a TB referral center over 6 years, was to evaluate the efficacy of this methodology in improving therapy guidance for clinicians and in improving the understanding of the epidemiology of TB transmission. WGS was performed in addition to pDST on 1001 strains consecutively isolated between January 2016 and December 2021; the results allowed us to improve the quality of data on resistance and to identify possible clusters of transmission. Prediction of rifampicin-resistant (RR) or multi-drug-resistant TB strains (MDR-TB, defined as resistance to at least rifampicin and isoniazid) was obtained for 50 strains (5%). Mutations predictive of an MDR isolate were further characterized, and Ser450Leu and Ser315Thr were found to be the most frequent mutations in *rpoB* and *katG* genes, respectively. Discordances between WGS and phenotypic drug susceptibility testing (pDST) were found in few strains, and their impact on clinical decisions and outcome was addressed. The introduction of WGS in our Institute improved our diagnostic routine, allowing accurate patient management, and was a valid instrument for epidemiological investigations and infection control.

## 1. Introduction

To this day, TB remains a public health priority in both low- and high-resources countries, representing one of the world’s leading causes of death from an infectious disease. Despite the slow downward trend evident since the year 2000, which is in line with the global targets and milestones set for the Sustainable Development Goals (SDGs) and WHO’s End TB Strategy, an estimated 10.6 million people fell ill with TB in 2022 [1]. A smaller decline compared to previous years was observed in 2020 and 2021, due to disruptions to diagnostic and treatment services during the COVID-19 pandemic; however, a major recovery in the number of people diagnosed and treated for TB was observed in 2022 [1].

TB control has been effective in reducing TB incidence and mortality in the European Union and European Economic Area (EU/EEA). Although the numbers and rates reported in 2021 should be interpreted with caution for the above-mentioned reasons, the ECDC [2,3] reported a notification rate of 7.4 per 100,000 population in 2021 in the area, overall and in the single countries. Likewise, also the number of reported rifampicin-resistant (RR) and MDR-TB cases continued to decline in EU/EEA countries, reaching a rate of 4.2% in 2021.

In line with the general trend, Italy (a low TB incidence country) has seen a reduction in the number of new cases during the last decade, reaching an incidence value of 4.2 cases per 100,000 population in 2021 [3]. Similar to other western European capital cities, Rome hosts an ethnically heterogeneous population with higher numbers of migrants from countries with high incidence of TB and MDR-TB. The consequence is that the rates observed in Rome are higher than those found in the country as a whole: data from 2020 indicate 6.6 cases of TB per 100,000 persons reported in Rome [4] vs. 3.8 cases per 100,000 in Italy as a whole [3].

The aim of further reducing these numbers globally has generated an enormous effort in implementing drug susceptibility strategies, and in developing new tests that enable a fast and accurate diagnosis of resistant TB. For many years, pDST has been the only method available for detecting drug resistance, and still is considered the reference standard. However, the technical and infrastructural requirements, and, above all, the long turnaround time, represent a serious diagnostic gap in terms of ease of use and clinical impact. More recently, PCR-based molecular methods and/or line probe assays were introduced in clinical practice, providing fast identification of selected resistance-associated mutations (especially if performed on the clinical sample) [5].

Furthermore, in 2020 and 2022 the WHO released updated treatment guidelines for drug-resistant TB that recommended an all-oral, Bedaquiline (BDQ)-based regimen for all MDR-TB cases (therefore dismissing injectable drugs), and the use of new or redesigned drugs for MDR-TB that carry additional resistance to Fluoroquinolones (FQN) [6,7]. These new regimens warrant a remodeling of commercial drug sensitivity tests that should be able to identify resistance to the new drugs and, potentially, to any future drug.

As a consequence, an increasing role has been conferred to WGS as an “All-in-One” system able to identify resistance-associated mutations within the whole genome. A great effort is currently being directed to building a database that associates genetic variants to drug resistance and clinical outcome. In the near future, this will hopefully allow the reliable detection of resistance to old and new drugs while reducing the burden on microbiology laboratories, if, at least for some drugs, pDST could be replaced by WGS as the reference standard [8,9,10,11,12].

The ‘L. Spallanzani’ National Institute for Infectious Diseases (INMI), where this study was conducted, is a referral hospital for TB and MDR-TB in the metropolitan area of Rome. WGS has been in use in our laboratory for research purposes since 2016, but was recently introduced also in the diagnostic algorithm.

The aim of our study, based on sequencing data obtained from 2016 to 2021, which included resistance genotypes, frequency of resistance-associated mutations, and cluster analysis, was to evaluate the efficacy of WGS in improving drug sensitivity testing and in aiding the clinician in the choice of the optimal therapy regimen.

## 2. Results

### 2.1. Study Population

One thousand and fifty-six MTB strains were collected during a 6-year period, from 2016 to 2021. All strains were identified by molecular methods and processed using pDST, according to the routine diagnostic procedure in use in the laboratory. Of these strains, 1025 (955 obtained from respiratory samples, 93%) were processed by WGS, and good quality sequences (suitable for bioinformatics analysis) were obtained for 1001 strains (Figure 1).

A total of 313 (31%) patients were born in Italy, and 688 (69%) were foreign-born. Only one strain per patient was included in the study. Multiple strains grown from different samples of the same patient were analysed only if treatment failure was suspected (which happened in three cases), and the results were included only if the pDST or WGS showed a different resistance profile.

Isolation and collection of strains was homogeneous during the study time, except for the years 2020 and 2021, in which we isolated a significantly lower number of MTB strains due to the impact of the COVID-19 pandemic, following the designation of our Institute as the main COVID hospital in the Latium Region [13].

### 2.2. Prediction of TB Drug Resistance Based on WGS

Analysis of whole genome sequences, carried out with the PhyResSE bioinformatic open tool, showed that 149 (15%) isolates harbored mutations predictive of resistance (any kind of resistance) to first- or second-line drugs (Table 1).

Fifty strains (5% of the total study population) were classified as RR or MDR. Specifically, according to the recently updated WHO definitions [6] we detected the following: 5 RR-TB, 34 MDR-TB, 10 pre-XDR-TB, and 1 XDR-TB strains.

Single resistance to INH was found in 38 of the non-MDR cases. Resistance to EMB, S, PZA, FQN, and injectable drugs was found in 4, 8, 6, 22, and 20 strains, respectively, all of which were otherwise susceptible to INH or RMP or both.

Sequences are available under the BioProject accession number: PRJNA834606.

### 2.3. Characteristics of the MDR Population

Further analysis of MDR-TB cases showed that eight (16%) were previously treated cases. Forty cases were isolated from foreign-born patients (80%). The rate of MDR-TB was 3.2% in the Italian-born population and 5.8% in the foreign-born. MDR patients were aged 19–82 years. Male–female ratio was 3.5.

### 2.4. Frequency of Mutations Predictive of RIF and INH Resistance in MDR Strains

Analysis of the mutations conferring resistance to RIF and INH in MDR-TB strains showed that more than 46% of the strains harboured the combination of Ser450Leu in the *rpoB* and Ser315Thr mutation in the *katG* genes. A further 17.8% had the same combination with an additional mutation in the *inhA* promoter in position −15, followed by the combination of Asp435Val and Ser315Thr (11%) (Table 2).

From a further analysis of the rpoB mutations identified in isolates with increasing level of resistance we found that Ser450Leu in the rpoB gene was well-represented in RR, MDR-TB, and pre-XDR isolates (40%, 69%, and 55%, respectively), followed by Asp435Val that was present in 33% of the pre-XDR-TB. The only XDR-TB isolate present in this study harbored the Ser450Leu in rpoB (Table 3).

### 2.5. Comparison of WGS and pDST

Concordance between prediction of resistance to RIF using WGS and pDST was found in 44/50 (88%) cases (Table 4). Six strains that were classified as susceptible using pDST were found to harbour mutations using WGS analysis: in four cases, the mutations detected in *rpoB* (Asp435Tyr and Leu430Pro) are known to be generally missed by rapid phenotypic DST methods [14,15,16,17]. The remaining two strains had Asp435Val and His445Tyr, respectively. These mutations are, according to the WHO catalogue of mutations [14], strongly associated with drug resistance and with a resistant pDST, but with low sensitivity.

For INH, we missed results from pDST for two strains, due to culture contamination; all other results showed concordance between the two methods. Several discrepancies between WGS and pDST were found for EMB and S. PZA testing with pDST was introduced in the laboratory in 2020, using a modified method [18].

For the new drugs BDQ and CFZ, currently used for the treatment of MDR-TB cases, a standardized database of mutations is not yet available. We found mutations in the *mmpR* gene (Rv0678), one of the genes associated with resistance to BDQ, in four MDR-TB strains. Phenotypic DST for BDQ was performed on three of these strains and resistance was found in all of them (not shown).

### 2.6. Cluster Analysis

Analysis of sequences using cgMLST revealed that 42% of the analysed strains were part of a cluster, considering a genetic distance of ≤four alleles as an indicator of recent transmission. Ninety-eight clusters were identified, with the highly transmissible lineage 4 (Euro–American) and the Haarlem sub-lineage being the most represented, followed by the East African Indian (Delhi/CAS) and East Asian (Beijing) lineages. Lineage 4 was consistently found in the clusters with the highest number of isolates (up to 38 strains).

Four of the identified clusters included RR/MDR strains carrying identical mutation profiles predictive of resistance to RIF. Three of them belonged to the Euro–American lineage and one to the East Asian (Beijing).

## 3. Discussion

The aim of this study was to evaluate the effect of the introduction of routine WGS on all isolated TB strains in a referral TB laboratory, in a setting of low TB incidence. The WGS technology had been used in our institute for research since 2016, but given our role as a regional reference centre we sought to introduce it into our routine laboratory practice, to provide clinicians with better quality data on antibiotic susceptibility.

To this end, we used pDST and WGS to study 1001 strains collected during a period of 6 years (Figure 1), to evaluate the impact of WGS in terms of added value to the standard pDST in our setting, while characterizing the MDR-TB population circulating in the greater metropolitan area of Rome at the molecular level, and possibly contributing to disease surveillance.

Our results showed a high level of concordance between WGS and pDST. We found that 15% of all MTB strains in our study carried at least one resistance-associated mutation, and that 5% were MDR-TB, which is higher than the national figure of 3.1% for 2021 [3] (Table 1). This is not surprising, partly because the metropolitan area of Rome is characterized by a highly heterogeneous population, with most of the foreign-born individuals coming from countries with high incidence of tuberculosis (up to 52.4/100.000 in some areas of Rome in 2019) [4]. When we analyzed the frequency of mutations conferring resistance to RIF and INH, we found that 46% of MDR-TB isolates carried the combination of Ser450Leu and Ser315Thr mutations in the *rpoB* and *kat*G genes, respectively (Table 2). Our findings are not surprising as all the mutations detected are well known and described [13]. No discordant results between pDST and WGS were found for PZA and INH. Several discrepancies were found for EMB and S, as widely described in literature [19].

We detected eight mutations associated with resistance to RIF in the *rpo*B gene, with Ser450Leu being present in 48% of MDR strains (Table 3). By comparing pDST and WGS results for RIF resistance, we found discordant results in 6/1001 strains (Table 4). These were cases of isolates carrying ‘disputed’ mutations, described in the literature as correlated with treatment failure despite a susceptible testing in vitro [13,14,15]. In four of those strains, the mutations detected were Asp435Tyr and Leu430Pro, while Asp435Val and His445Tyr were found in the remaining two. All those mutations are known for being generally missed by rapid phenotypic DST methods (Asp435Tyr and Leu430Pro) [14,15,16,17], or for being detectable with low sensitivity in the pDST (Asp435Val and His445Tyr) [14].

We received positive feedback from clinicians, who were, in some cases, able to re-evaluate and adjust individual treatment regimens in selected cases. As a result, we decided to include this methodology into our diagnostic algorithm (Figure 2), and to produce an official laboratory report for the clinical department, according to Institute guidelines [20].

The idea of replacing pDST with WGS, at least for first-line drugs, is very appealing, and several studies are ongoing with the aim of building evidence for using this method as the primary DST [21,22,23]. In our study, 951 strains were classified as susceptible by both pDST and WGS, and in no case did a ‘wild-type’ genome display a resistant phenotype by pDST (Table 4). Our results suggest that eliminating the need of pDST for strains that show the absence of resistance-correlated mutations using WGS could be considered in the future.

Several studies explored the use of a WGS-based approach for tuberculosis surveillance, evaluating for example the possibility of creating centralized WGS-based surveillance systems involving large geographical areas. These studies showed that this approach could also efficiently elucidate the dynamics of in-country and cross-border RR/MDR-TB transmission [24,25]. We found a considerable number of close clusters (i.e., including strains that are ≤four alleles apart, which is suggestive of recent transmission) with 41.6% of total strains found to belong to a cluster. Most of these included isolates of the modern Euro–American lineage, with the Haarlem sub-lineage being represented the most. Of particular note is our identification of four MDR-TB clusters that highlights the occurrence of transmission of resistant strains.

This study was conducted on a quite large population (1001 strains), and although it was an acceptable representation of the wider geographical region because of the role of our Institute as referral centre, all strains were collected only in our hospital and this constitutes a limitation.

Further research is needed to standardize WGS for new drugs, such as BDQ and CFZ, which were recently introduced in the new treatment guidelines [5,6,26,27]. In our laboratory we adopted a mixed approach for these new drugs in agreement with the available literature, performing both WGS and pDST [28].

Despite these limitations, implementing WGS in clinical practice allowed us to improve the diagnosis of tuberculosis in our hospital. WGS results are now taken into consideration for clinical decision-making to confirm treatment choices or modify drugs already in use in case of discrepancies or the occurrence of side effects.

The possibility of completely relying on WGS for the diagnosis and DST of MTB has still several limitations, the most important being the long turnaround time caused by the need of a culture. Further research and validation studies are needed for the development of WGS on the primary sample, which would represent a key step towards the successful treatment of MDR-TB.

## 4. Materials and Methods

### 4.1. Study Population and Setting

This retrospective/observational study included all MTB strains consecutively isolated in our laboratory from pulmonary and extra-pulmonary samples between January 2016 and December 2021 from patients with a microbiologically confirmed diagnosis of tuberculosis. Samples were subjected to diagnostic routine testing, according to national and local guidelines [20].

All strains were isolated from samples collected from the wider Latium area, in line with the role of the L. Spallanzani Institute as a TB referral hospital. Results from microbiological and molecular tests were collected from the records of the laboratory of Microbiology, and linked to demographic data for analysis. Demographic data included age, sex, and whether the patients were born in Italy or in a foreign country. Information on previous TB episodes was collected for MDR-TB cases.

### 4.2. Diagnosis and Drug Susceptibility Testing

The laboratory diagnostic workflow included smear microscopy together with culture and identification by a molecular method performed in parallel on the primary sample. The following molecular tests are available in our laboratory: Xpert MTB/RIF (Cepheid, Sunnyvale, CA, USA), GenoType MTBDR plus/sl (Hain Lifescience, GmbH, Nehren, Germany), Anyplex^TM^ II MTB/MDR, MTB/XDR Detection (Seegene Inc., Seul, Republic of Korea) and BD MAX™ MDR-TB (Becton Dickinson, Franklin Lakes, NJ, USA). Liquid culture of samples for rapid automated cultivation (Mycobacteria Growth Indicator Tubes-MGIT, incubated in MGIT 960; Becton Dickinson, MD, USA) was performed on all samples.

Automated pDST for first-line drugs was performed using the MGIT automated system (critical concentrations: RIF 1.0 or 0.5 mg/L, following WHO recommendations at the time of testing [29]; INH 0.1 and 0.4 mg/L; EMB 5.0 and 7.5 mg/L and PZA 100 mg/L). The pDST for second-line drugs (MFX, LFX, AMK, and more recently BDQ and LZD) was performed using a semi-automated system (BD EpiCenter™ TB-eXiST).

### 4.3. Whole Genomic Sequencing

Whole genomic sequencing (WGS) was performed on genomic DNA extracted from isolated strains, initially as part of an in-house method evaluation and then increasingly used as support to the clinicians for accurate therapy design.

Sequencing was carried out using the Ion Torrent or Illumina systems. Briefly, the Ion Xpress Plus Fragment Library Kit (ThermoFisher Scientific, Waltham, MA, USA) was used to prepare sample libraries, and the Chef and S5 platforms were used for chip preparation and sequencing, to obtain 250 bp reads. Alternatively, 150 bp or 300 bp paired reads were generated using Nextera XT DNA Library Prep Kit (Illumina Inc., San Diego, CA, USA) and NextSeq 500 or MiSeq sequencers, respectively (Illumina Inc.). The reference genome breadth was ≥92% with a mean depth of coverage of ≥30×.

### 4.4. Sequence Analysis for Prediction of Drug Resistance and Cluster Identification

Sequence reads obtained with WGS were submitted to the PhyResSE pipeline [30] in order to predict susceptibility or resistance based on the identification of mutations associated with drug resistance.

Mutations identified in the genes *rpoB*, *katG*, and *inhA-promoter* were further characterized in MDR-TB strains, and frequencies were calculated.

Reads were also submitted to the SeqSphere+ software (RIDOM bioinformatics GmbH, Munster, Germany) for cgMLST (core genome MLST) analysis and identification of clusters. Strains harbouring mutations associated with an MDR/RR status were further analysed in order to identify possible clustering and transmission of resistant strains.

### 4.5. Ethics

This analysis was conducted in the context of an observational study on tuberculosis (DB/TB) approved by the Ethics Committee at INMI (decision n. 12/2015).

Strains and demographic data were collected in accordance with the Ethics committee of INMI, and in our capacity of regional reference and surveillance center.

## Figures and Tables

**Figure 1 antibiotics-13-00134-f001:**
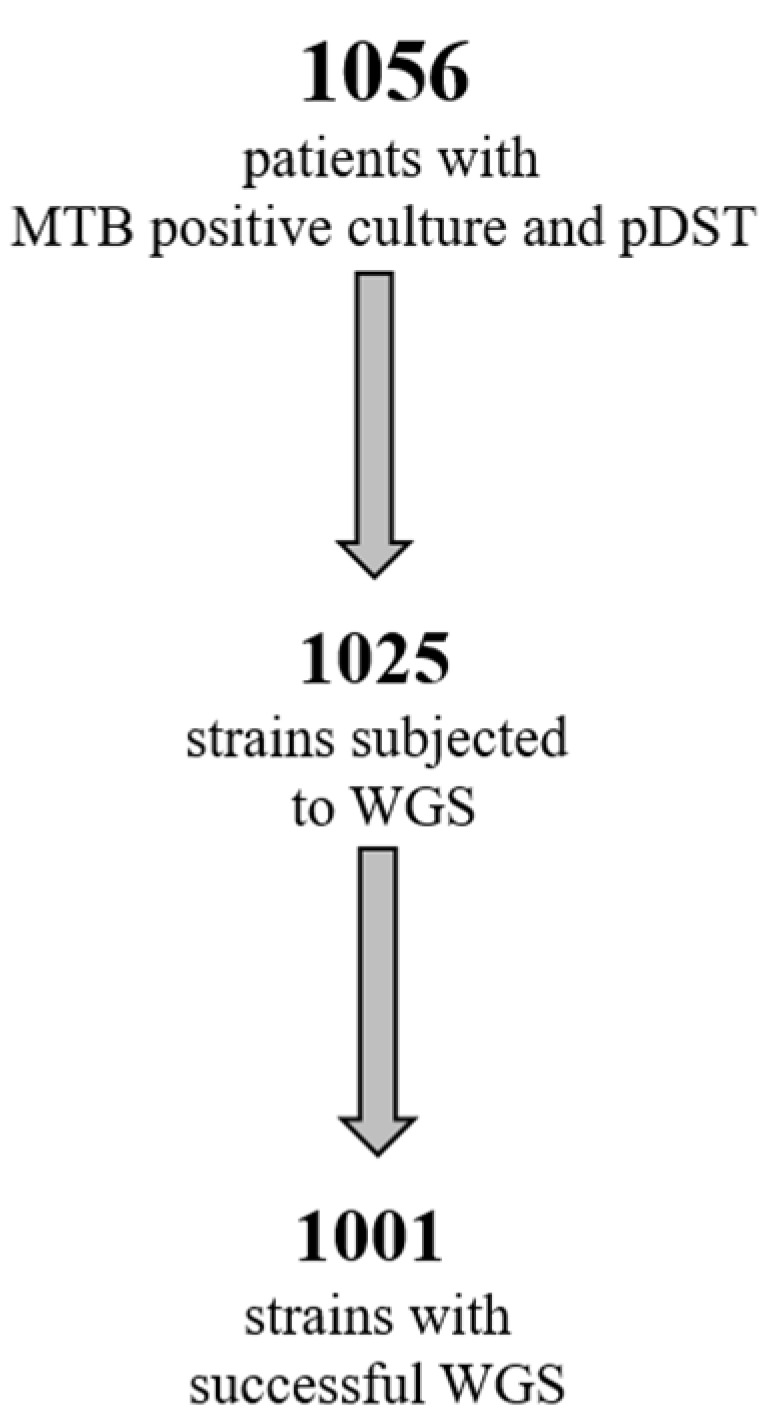
Number of MTB strains available for pDST and WGS.

**Figure 2 antibiotics-13-00134-f002:**
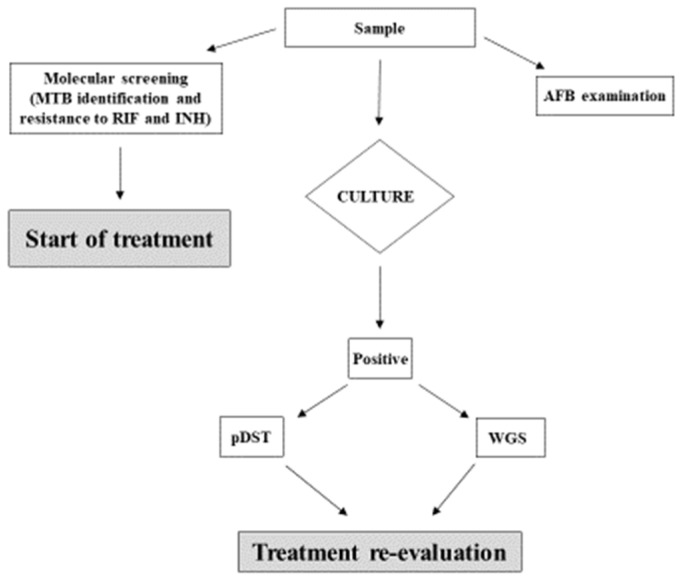
Diagnostic algorithm for TB used at the INMI Institute. The clinical sample undergoes microscopic examination of acid-fast bacilli (AFB), traditional culture, and first-line molecular screening for MTB and INH resistance. Results of molecular biology testing performed on primary samples are taken into consideration in case of discordance with DST on the isolate.

**Table 1 antibiotics-13-00134-t001:** Anti-tuberculosis drug resistance based on WGS results, in the studied population.

Sensitive N. (%)	Resistant N. (%)	Single Resistance, Not MDR N. Cases (%)						RR/MDR N. Cases (%)			
		INH	EMB	S	PZA	FQN	I.D.	RR	MDR	Pre-XDR	XDR
852 (85)	149 (15)	38 (3.8)	4 (0.4)	8 (0.8)	6 (0.6)	22 (2.2)	20 (2.0)	5 (0.5)	34 (3.4)	10 (1.0)	1 (0.1)
Total N. 1001 (100)		Total N. 99 (10)						Total N. 50 (5)			

Sensitive: drug-susceptible *Mycobacterium tuberculosis*; INH: isoniazid; EMB: ethambutol; S: streptomycin; PZA: pyrazinamide; FQN: fluoroquinolones; I.D.: injectable drugs (amikacin, kanamycin and capreomycin); RR: resistance to rifampicin; MDR (multi-drug resistance): resistant to at least isoniazid and rifampicin; Pre-XDR (pre-extensively drug-resistant): MDR plus an additional resistance to fluoroquinolones; XDR (extensively drug-resistant): MDR plus resistance to any fluoroquinolone and at least one additional Group A drug (bedaquiline and/or linezolid). Classification of drug-resistant strains is based on WHO new definitions (Report 27–29 October 2020).

**Table 2 antibiotics-13-00134-t002:** Frequency of mutations associated with resistance to RIF (in the *rpoB* gene) and isoniazid (Ser315Thr in the *katG* gene and position −15 in the promoter of the *inhA* gene) in MDR-TB strains.

	RIF-R N. (%)	Leu452Pro	Ser450Leu	Asp435Val	Gln432Pro	Leu430Pro	His445Tyr	Asp435Tyr	Total
									N. (%)
**INH-R** **N. (%)**	Ser315Thr	1 (2.22)	21 (46.7)	5 (11.1)	2 (4.42)	1 (2.22)	1 (2.22)	3 (6.7)	34 (75.56)
	*inhA-prom* −15		1 (2.22)						1 (2.22)
	Ser315Thr + *inhA prom*		8 (17.8)				2 (4.44)		10 (22.2)
Total N. (%)		1 (2.22)	30 (66.7)	5 (11.1)	2 (4.42)	1 (2.22)	3 (6.7)	3 (6.7)	45 (100)

**Table 3 antibiotics-13-00134-t003:** Distribution of the most frequent mutations in *rpoB* associated with resistance to rifampicin in relation to the level of resistance.

Level of Resistance					
*rpoB* Mutation N. (%)	RIF-R	MDR	Pre-XDR	XDR	Total N. (%)
Leu452Pro			1 (2)		1 (2)
Ser450Leu	2 (4)	24 (48)	5 (10)	1 (2)	32 (64)
Asp435Val		2 (4)	3 (6)		5 (10)
Gln432Pro		2 (4)			2 (4)
Leu430Pro	1 (2)	1 (2)			2 (4)
His445Tyr	1 (2)	3 (6)			4 (8)
Asp435Tyr		3 (6)			3 (6)
His445Asp	1 (2)				1 (2)
Total N. (%)	5 (10)	35 (70)	9 (18)	1 (2)	50 (100)

**Table 4 antibiotics-13-00134-t004:** Comparison of pDST and WGS results obtained on 1001 MTB strains.

	**pDST**			
**WGS**		S	R	Total
	S	951	0	951
	R	6	44	50
	Total	957	44	1001

## Data Availability

Sequences are available under the BioProject accession number PRJNA834606. All strains included in this study were stored at the Institute Biobank and are available for further studies.

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
