# Peer review of "Implementation of Whole Genome Sequencing of Tuberculosis Isolates in a Referral Center in Rome: Six Years’ Experience in Characterizing Drug-Resistant TB and Disease Transmission"

_antibiotics, 2024, doi:10.3390/antibiotics13020134_

Round 1

Reviewer 1 Report

Comments and Suggestions for Authors

In the article titled "Implementation of Whole Genome Sequencing of Tuberculosis Isolates in a Referral Center in Rome: Six Years' Experience in Characterizing Drug-Resistant TB and Disease Transmission" by Cannas et al., the authors conducted a comprehensive study on the whole genome sequencing strains (isolates) of Mycobacterium tuberculosis. These isolates were purified over a span of six years, from 2016 to 2021. The primary focus of the article pertains to drug-resistant tuberculosis and its associated implications. Nevertheless, the clarity of the manuscript is compromised, rendering some of the authors' statements challenging to comprehend. The article is poorly written and needs more efforts.

Several noteworthy concerns are outlined below:

Major Issues:

The manuscript lacks clarity on whether the specific strains were subjected to 16S rRNA sequencing for identification before undergoing whole genome sequencing.

Lines 83 to 85 are perplexing; clarification is needed regarding whether the strains were sequenced prior to their inclusion in the study.

The abbreviations employed throughout the manuscript require greater precision, particularly concerning terms such as INH-R, EMB-R, SM-R, PX-R, FQ-R, and ID-R.

Even in the abstract no abbreviations are mentioned.

Distinctions between Pre-XDR and MDR warrant elucidation.

The manuscript should include a concise mention of the total number of isolates, the quantity subjected to sequencing, and the number analyzed.

The discussion section does not sufficiently address the potential variability in resistance patterns among isolates, even if originating from different patients but of the same strain.

Minor Issues:

The article necessitates a linguistic refinement to enhance clarity.

Future plans for the isolates, such as alterations to treatment regimens based on study outcomes, are not articulated and should be addressed.

In summary, the aforementioned concerns highlight critical and ancillary issues within the manuscript, warranting attention and revision for enhanced clarity and scientific rigor.

Comments on the Quality of English Language

The article requires extensive editing for the english language and may be a sr author needs to read the MS before it is resubmitted. The english and the presentation is very novice

Author Response

Reviewer 1

In the article titled "Implementation of Whole Genome Sequencing of Tuberculosis Isolates in a Referral Center in Rome: Six Years' Experience in Characterizing Drug-Resistant TB and Disease Transmission" by Cannas et al., the authors conducted a comprehensive study on the whole genome sequencing strains (isolates) of Mycobacterium tuberculosis. These isolates were purified over a span of six years, from 2016 to 2021. The primary focus of the article pertains to drug-resistant tuberculosis and its associated implications. Nevertheless, the clarity of the manuscript is compromised, rendering some of the authors' statements challenging to comprehend. The article is poorly written and needs more efforts.

Several noteworthy concerns are outlined below:

Major Issues:

The manuscript lacks clarity on whether the specific strains were subjected to 16S rRNA sequencing for identification before undergoing whole genome sequencing.

Reply: we have specified in the manuscript that all strains studied had been identified by molecular methods (Page 2 lines 86-90, and Page 8, lines 282-284 for details on molecular tests available at the laboratory)

Lines 83 to 85 are perplexing; clarification is needed regarding whether the strains were sequenced prior to their inclusion in the study.

Reply: we have corrected the paragraph in question: only one strain per patient was subjected to sequencing, unless we received a specific request by the clinicians, as it happened for three patients that were known for their non-compliance to therapy (Page 3, Lines 98-101).

The abbreviations employed throughout the manuscript require greater precision, particularly concerning terms such as INH-R, EMB-R, SM-R, PX-R, FQ-R, and ID-R.

Reply: we have checked and corrected the abbreviations throughout

Even in the abstract no abbreviations are mentioned.

Reply: we have added abbreviations in the abstract

Distinctions between Pre-XDR and MDR warrant elucidation.

Reply: we have clarified the definitions of MDR and Pre-XDR as per current WHO definitions (Page 1, Lines 18-20, Page 3 Lines 114-116)

The manuscript should include a concise mention of the total number of isolates, the quantity subjected to sequencing, and the number analyzed.

Reply: we have included this information in the text (Page 2, Lines 87-90) and also added Figure 1 to better clarify.

The discussion section does not sufficiently address the potential variability in resistance patterns among isolates, even if originating from different patients but of the same strain.

Reply: we have completely re-written the discussion, including paragraphs on the WGS results (Page 6, lines 196-216)

Minor Issues:

The article necessitates a linguistic refinement to enhance clarity.

Reply: the article was extensively revised and re-written.

Future plans for the isolates, such as alterations to treatment regimens based on study outcomes, are not articulated and should be addressed.

Reply: We added information on the clinical outcome of WGS, i.e. how the clinicians interpret and use the results, and added Figure 2 that shows the diagnostic flow used in our laboratory (Page 6 Lines 217-221, and Figure 2)

In summary, the aforementioned concerns highlight critical and ancillary issues within the manuscript, warranting attention and revision for enhanced clarity and scientific rigor

Reviewer 2 Report

Comments and Suggestions for Authors

The authors have investigated a WGS approach to characterisation of drug-resistance in TB and disease transmission. The manuscript is constructed in a logical manner however inconsistencies are observed within the formatting and text usage.

Line 63                  change tuberculosis to TB

Line 79                  change tuberculosis to TB

Line 85                  The change in profile between the initial test and final isolates is significant and should have been explored in more depth.

Line 93                  Percentage Romanian? Significance in the final analysis, this wasn’t mentioned further.

Line 98                  15% resistance rate compared to the 6.6% stated in the introduction. This is significantly higher. Why is there such a vast difference?

Line 194               “Isoniazid” remove the e or change to INH as abbreviated above.

Line 195               “Streptomycin”

Line 249-252      The reviewer is not convinced that this is a method which will speed up therapy implementation. However, the significance of identify RIF resistance outside the usual XPERT-RIF test and the identification of other mutations is useful for diagnosis and treatment. Using WGS and bioinformatic profiling to influence treatment is time consuming and the current availability is not sufficient to propose a pharmacogenomic approach to TB treatment.  

Comments on the Quality of English Language

The authors need to decide if they’re writing the full name or using abbreviations. This needs to be consistent throughout the document. e.g. MTB not defined, tuberculosis vs TB etc. Especially with the drug names, also RIF is the more common annotation for rifampicin

Author Response

Reviewer 2

The authors have investigated a WGS approach to characterisation of drug-resistance in TB and disease transmission. The manuscript is constructed in a logical manner however inconsistencies are observed within the formatting and text usage.

Line 63 change tuberculosis to TB

Reply: tuberculosis has been changed to TB throughout the manuscript

Line 79 change tuberculosis to TB

Reply: tuberculosis has been changed to TB throughout the manuscript

Line 85 The change in profile between the initial test and final isolates is significant and should have been explored in more depth.

Reply: we have clarified that the 3 cases that showed a change in their resistance profile were patients known for their lack of compliance to therapy, so multiple strains were analysed because treatment failure was suspected (Page 3 Lines 98-101)

Line 93 Percentage Romanian? Significance in the final analysis, this wasn’t mentioned further.

Reply: We have deleted the sentence, as it is not relevant for the study conclusions, as the reviewer correctly pointed out.

Line 98 15% resistance rate compared to the 6.6% stated in the introduction. This is significantly higher. Why is there such a vast difference?

Reply: we have clarified in the text that 6.6 is the incidence of TB/100.000 population in the metropolitan area of Rome (Page 2 Line 51), while 15% is the percentage of strains harbouring any resistance (single resistance 10% + RR/MDR 5%) found in our study (Table 1, Page 3)

Line 194 “Isoniazid” remove the e or change to INH as abbreviated above.

Reply: Abbreviations are used throughout the manuscript

Line 195 “Streptomycin”

Reply: Abbreviations are used throughout the manuscript

Line 249-252 The reviewer is not convinced that this is a method which will speed up therapy implementation. However, the significance of identify RIF resistance outside the usual XPERT-RIF test and the identification of other mutations is useful for diagnosis and treatment. Using WGS and bioinformatic profiling to influence treatment is time consuming and the current availability is not sufficient to propose a pharmacogenomic approach to TB treatment.

Reply: we agree with the reviewer and modified the discussion, highlighting other useful implications for patient management (Page 6 Lines 208-221 also the newly added Figure 2 on page 6)

Comments on the Quality of English Language

The authors need to decide if they’re writing the full name or using abbreviations. This needs to be consistent throughout the document. e.g. MTB not defined, tuberculosis vs TB etc. Especially with the drug names, also RIF is the more common annotation for rifampicin

Reply: we have corrected the abbreviations throughout the manuscript

Reviewer 3 Report

Comments and Suggestions for Authors

I appreciate the opportunity to review the manuscript entitled "Implementation of Whole genome sequencing of tuberculosis 2 isolates in a referral center in Rome: six years’ experience in 3 characterizing drug-resistant TB and disease transmission". In the current prospective study by Angela Cannas et al. have evaluated the use of whole genome sequencing method for the diagnosis of tuberculosis from TB isolates collected in a referral center in Rome over 6 years. 

However, the author may take note of the major and minor remarks listed below to improve the manuscript:

Major comments:

  • The manuscript is poorly written and requires extensive corrections for redundancy, grammatical errors, consistency, and wordiness.
  • The introduction needs to be completely rewritten as it is randomly written and lacks relevance and justification to the problem of study or study hypothesis. I request that authors include epidemiology, both global and national, drug resistance, conventional diagnostic methods, challenges with conventional methods and the importance or ease of WGS when it is implemented.
  • Please emphasize the problem of the study based on earlier data.
  • Line No. 82–83: Please clear the message in this line.
  • line no. 93: Please justify ‘what is the context of adding this line here’.
  • It would be very useful if the author included the comparative data of pDST.
  • I would highly encourage authors to include results as figures to make them more informative.
  • The discussion is poorly written and needs to be rewritten completely. It should be structured based on the observed results.
  • Also, if authors want to emphasize the ethnicity and origin of the country, please provide these numbers in the results section.
  • Additionally, the study duration is quite old; please clarify this point.
Comments on the Quality of English Language
  • The manuscript is poorly written and requires extensive corrections for redundancy, grammatical errors, consistency, and wordiness.

Author Response

Reviewer 3

I appreciate the opportunity to review the manuscript entitled "Implementation of Whole genome sequencing of tuberculosis 2 isolates in a referral center in Rome: six years’ experience in 3 characterizing drug-resistant TB and disease transmission". In the current prospective study by Angela Cannas et al. have evaluated the use of whole genome sequencing method for the diagnosis of tuberculosis from TB isolates collected in a referral center in Rome over 6 years.

However, the author may take note of the major and minor remarks listed below to improve the manuscript:

Major comments:

  • The manuscript is poorly written and requires extensive corrections for redundancy, grammatical errors, consistency, and wordiness.

Reply: we have extensively re-written the manuscript

  • The introduction needs to be completely rewritten as it is randomly written and lacks relevance and justification to the problem of study or study hypothesis. I request that authors include epidemiology, both global and national, drug resistance, conventional diagnostic methods, challenges with conventional methods and the importance or ease of WGS when it is implemented.

Reply: we have re-written the introduction to make it more focused towards the aim of the study (Pages 1 and 2)

  • Please emphasize the problem of the study based on earlier data.

Reply: we have clarified the aim of the study, which was to describe how WGS was implemented in our diagnostic flow and how the results can translate into a better patient management (Page 2 Lines 71-74 and 79-82)

  • Line No. 82–83: Please clear the message in this line.

Reply: we have modified the text to clarify how molecular tests were used and added more information in general on how strains were selected (Page 2, lines 86-90, Page 3, Lines 98-101, Page 8, Lines 280-287). We also added Figure 1 (Page 3) and Figure 2 (Page 6) to better clarify the strain selection process and how the WGS was included in our diagnostic algorithm, respectively.

  • line no. 93: Please justify ‘what is the context of adding this line here’.

Reply: the reviewer is correct, the line is out of context and we removed it.

  • It would be very useful if the author included the comparative data of pDST.

Reply: we included Table 4 to graphically show the comparison of WGS and pDST data (Page 5), and discussed the results (Page 5, Lines 151-157 and Page 7, Lines 232-236).

  • I would highly encourage authors to include results as figures to make them more informative.

Reply: this was a very good suggestion, we added Figures 1 and 2, and Tables 4 (Pages 3, 6, and 5, respectively).

  • The discussion is poorly written and needs to be rewritten completely. It should be structured based on the observed results.

Reply: we have re-written the discussion to logically follow the results

  • Also, if authors want to emphasize the ethnicity and origin of the country, please provide these numbers in the results section.

Reply: we have deleted the information on ethnicity, as it is not relevant for the aim and conclusions of the study, as suggested by the reviewer.

  • Additionally, the study duration is quite old; please clarify this point.

Reply: the reviewer is correct in pointing out that the duration of the study is quite old. This is largely due to the Covid pandemic during which our hospital (which is an infectious diseases hospital with ICU capacity) was converted into the main Covid hospital of the city of Rome. The consequence was that the patient population largely changed and also the hospital and laboratory staff became more heavily involved in other activities. We have included a short sentence to comment on this (Page 3, Lines 98-101). We also added reference 13.

  • Comments on the Quality of English Language The manuscript is poorly written and requires extensive corrections for redundancy, grammatical errors, consistency, and wordiness.

Reply: the manuscript has been extensively re-written

Round 2

Reviewer 1 Report

Comments and Suggestions for Authors

Authors has made all the appropriate changes what I asked in the first review.

Comments on the Quality of English Language

There are some minor english language concerns, I would appreciate if the MS is properly edited by someone expert in english language. 

Reviewer 3 Report

Comments and Suggestions for Authors

Thank you for adding necessary changes